# Insect Peptide CopA3 Mitigates the Effects of Heat Stress on Porcine Muscle Satellite Cells

**DOI:** 10.3390/ani13203209

**Published:** 2023-10-14

**Authors:** Jeongeun Lee, Shah Ahmed Belal, Xi Lin, Jinryong Park, Kwanseob Shim

**Affiliations:** 1Department of Agricultural Convergence Technology, Jeonbuk National University, Jeonju 54896, Republic of Korea; dlwjddms0625@naver.com; 2Department of Animal Biotechnology, Jeonbuk National University, Jeonju 54896, Republic of Korea; sabelal.sau@gmail.com; 3Department of Animal Science, North Carolina State University, Raleigh, NC 27695, USA; xilin@ncsu.edu; 4Department of Stem Cell and Regenerative Biotechnology, Konkuk University, Seoul 05029, Republic of Korea; 53D Tissue Culture Research Center, Konkuk University, Seoul 05029, Republic of Korea

**Keywords:** porcine satellite cell, CopA3, cell viability, heat shock protein, heat stress

## Abstract

**Simple Summary:**

In this study, our objective was to assess the efficacy of incorporating insect peptide CopA3 to stabilize the proliferation and heat shock protein (HSP) expression under varying temperatures (37, 39, and 41 °C) of porcine muscle satellite cells (PMSCs). We observed that treatment with CopA3 mitigated the deleterious impact of heat stress on cell viability. Furthermore, CopA3 exhibited a reduction in PMSCs apoptosis, while stabilizing the expression of Bcl-2-associated X (BAX), HSP70, and HSP40. These results suggest the potential of CopA3 as a heat-stress alleviator.

**Abstract:**

Heat stress inhibits cell proliferation as well as animal production. Here, we aimed to demonstrate that 9-mer disulfide dimer peptide (CopA3) supplementation stabilizes porcine muscle satellite cell (PMSC) proliferation and heat shock protein (HSP) expression at different temperatures. Therefore, we investigated the beneficial effects of CopA3 on PMSCs at three different temperatures (37, 39, and 41 °C). Based on temperature and CopA3 treatment, PMSCs were divided into six different groups including treatment and control groups for each temperature. Cell viability was highest with 10 µg/mL CopA3 and decreased as the concentration increased in a dose-dependent manner. CopA3 significantly increased the cell viability at all temperatures at 24 and 48 h. It significantly decreased apoptosis compared to that in the untreated groups. In addition, it decreased the apoptosis-related protein, Bcl-2-associated X (BAX), expression at 41 °C. Notably, temperature and CopA3 had no effects on the apoptosis-related protein, caspase 3. Expression levels of HSP40, HSP70, and HSP90 were significantly upregulated, whereas those of HSP47 and HSP60 were not affected by temperature changes. Except HSP90, CopA3 did not cause temperature-dependent changes in protein expression. Therefore, CopA3 promotes cell proliferation, inhibits apoptosis, and maintains stable HSP expression, thereby enhancing the heat-stress-tolerance capacity of PMSCs.

## 1. Introduction

Accelerated global warming is the main cause of severe economic losses and animal welfare problems in the livestock industry [1]. Heat stress refers to the physiological strain that occurs when cells or organisms are exposed to high temperatures [2]. It disrupts normal physiological and molecular functions [2] and protein stability [3], eventually leading to the loss of cell function [4], which negatively affects an animal’s performance [5]. Furthermore, heat stress reduces feed intake, which decreases muscle growth, and induces oxidative stress, apoptosis, and inflammatory signaling in the skeletal muscles of pigs [6,7,8,9,10]. In livestock, skeletal muscle is the key component of lean body mass, whose growth significantly affects overall body growth [11]. Skeletal muscle satellite cells, a heterogeneous population of adult stem cells present between the muscle sarcolemma and basal lamina of muscle fibers, are responsible for skeletal muscle regeneration and mass [12]. Heat stress inhibits porcine muscle satellite cell (PMSC) proliferation by regulating the cell cycle [13]. Therefore, the development of heat-stress-alleviating materials for PMSCs is crucial to ensure the sustainability of the livestock industry.

The insect antimicrobial peptide CopA3 is a 9-mer disulfide dimer peptide (LLCIALRKK-NH2) synthesized from coprisin, which is extracted from the Korean dung beetle (*Copris tripartitus*) [14]. CopA3 exerts antimicrobial, anti-inflammatory, anticancer, and anti-neuronal apoptotic effects [14,15,16,17]. Moreover, it prevents neuronal cell apoptosis by degrading p27Ki1 [14]. Addition of the insect antimicrobial peptide, CopA3, to the medium decreases the oxidative stress in chicken skeletal muscle fibroblasts under heat-stress conditions [18].

Several reviews have highlighted the nutritional (diet) approaches that can be used to overcome heat stress in livestock [19,20]. The proliferation of muscle cells is closely related to the muscle mass and growth rate of an animal. Hu et al. [21] reported that the addition of antimicrobial peptides to the diet of broilers exposed to heat stress resulted in the restoration of body weight to non-heat stressed chicken. Therefore, in this study, we investigated whether antimicrobial peptide CopA3 restores the proliferation of heat-stressed PMSCs in vitro. Our results can aid in the development of heat-stress-alleviating materials.

## 2. Materials and Methods

### 2.1. Ethical Statement

This study was approved by the Animal Ethics Committee of Jeonbuk National University, Republic of Korea (approval number: JBNU 2020-0147). All experiments were performed according to the guidelines and regulations of Jeonbuk National University.

### 2.2. Peptide Synthesis

CopA3 was synthesized by AnyGen Co., Ltd. (Gwangju, Republic of Korea). The peptide was purified via reverse-phase high-performance liquid chromatography using the Inspire 5 μm C18 column (Dikma Technologies, Lake Forest, CA, USA), and its final purity was 98.1%. CopA3 in powder form was dissolved in distilled water and stored at −20 °C.

### 2.3. PMSC Isolation

PMSCs were isolated from 1-day-old male piglets, as previously described [13]. Briefly, muscle tissues were collected from biceps femoris and washed using Dulbecco’s phosphate-buffered saline (Gibco, Carlsbad, CA, USA) supplemented with 1% penicillin–streptomycin (PS; Gibco). Subsequently, the fascia and fat were removed, and the tissue was chopped using scissors. The tissues were dissociated with 1 U/mL of dispase II (Roche, Indianapolis, IN, USA), 2 mg/mL of collagenase D (Roche), 0.25% trypsin-EDTA (Gibco), and 10% PS in Dulbecco’s modified Eagle’s medium/nutrient mixture F-12 (DMEM/F12; Gibco, Carlsbad, CA, USA) at 37 °C for 1 h. After digestion, the tissue was first filtered through a 100 µm cell strainer and then through a 40 µm cell strainer. After centrifugation at 204× *g* for 5 min, the supernatant was removed and the Ammonium–Chloride–Potassium lysis buffer was added to the cell pellet for 5 min at 4 °C. The cells were then centrifuged at 204× *g* for 5 min and resuspended with 15% fetal bovine serum (FBS; Gibco) and 1% PS–glutamine (PSG; Gibco) in DMEM/F12. Then, we used the pre-plating methods to isolated the satellite cells [13]. Briefly, the cells were seeded in a 0.1% gelatin-coated dish. After 1 h, suspended cells in the dish were collected to separate the satellite cells and fibroblasts and transferred to another dish.

### 2.4. Immunocytochemistry

The PMSCs and fibroblasts were seeded in a 4-well plate at a density of 5 × 10^4^ and cultured until they reached at 70–80% confluency. The cells were fixed by 4% paraformaldehyde at 4 °C for 20 min. After then, the cells were washed with phosphate-buffered saline (PBS) three times and incubated with blocking solution (PBS containing 3% BSA and 0.3% triton X-100) at room temperature for 2 h. After blocking, the cells were washed three times with washing solution (PBS containing 0.3% Triton X-100) and incubated overnight at 4 °C with primary antibodies. The primary antibodies used were anti-PAX7 (1:20, DSHB, Iowa, IA, USA) and anti-MYOD (1:200, Proteintech, Rosemont, IL, USA). Afterward, the cells were washed three times with washing solution, and then the secondary antibodies (Alexa Flour-488 or 568, 1:1000, Molecular probes, Eugene, OR, USA) were added to detect the fluorescent label at room temperature for 2 h. To visualize the nuclei, the 4′-6-diamidino-2-phenylinodole (DAPI) was stained for 5 min at room temperature. Fluorescence images were acquired using Leica 9900. The number of PAX7^+^/DAPI^+^ and MYOD^+^/DAPI^+^ cells were expressed as a percentage of the total nuclei number of cells counted, based on DAPI staining.

### 2.5. Experimental Design and Cell Culture Conditions

In this study, we designed the experiment to include six groups, consisting of both CopA3-treated (CopA3^+^) and untreated (CopA3^–^) groups at three temperature conditions (37, 39, and 41 °C). PMSCs were cultured in a proliferation medium (PM; DMEM/F12 medium containing 15% FBS and 1% PSG) and maintained in a 5% CO_2_ incubator at 37 °C. After stabilization for 24 h, cell culture plates were moved to each temperature incubator and cultured for 48 h to induce heat stress. The culture medium was changed every 24 h.

### 2.6. Cell Viability Assay

The viability of PMSCs was assessed using the cell counting kit-8 (CCK-8; Dojindo, Kumamoto, Japan). To establish the optimal concentration of CopA3, CopA3-untreated groups were maintained in the PM, whereas the culture medium of CopA3-treated groups was supplemented with 5, 10, 25, and 50 μg/mL of CopA3 at 37 °C. The final concentration of CopA3 was selected based on the cell viability assay results. After selecting an appropriate concentration of CopA3, cell viability was measured at 37, 39, and 41 °C. Briefly, cells (4 × 10^3^ cells/well) were seeded in a 96-well plate and stabilized at 37 °C for 24 h. After stabilization, the culture media of the CopA3-treated groups were substituted with the CopA3-supplemented media, and each culture plate was transferred to a separate incubator at three different temperatures (37, 39, and 41 °C). Cell viability was assessed 24 and 48 h after CopA3 treatment and exposure to heat stress, according to the manufacturer’s guidelines. Optical density was measured at 450 nm using a microplate reader (Thermo Fisher Scientific, Waltham, MA, USA).

### 2.7. Flow Cytometry Analysis

Cells were seeded in a 6 cm^2^ dish at a density of 3 × 10^5^ cells/dish and cultured. After cultivation for 72 h, the cells were trypsinized and washed with cold PBS containing 1% bovine serum albumin (Sigma-Aldrich, St. Louis, MO, USA). For cell cycle analysis, the cell pellet was fixed with 70% ethanol for 5 min at 4 °C. After centrifugation at 850× *g* for 5 min, 0.1 mg/mL RNase (Sigma-Aldrich) and propidium iodide (PI) solution (Bio Legend, San Diego, CA, USA) were added to the cell pellet. For apoptosis analysis, the FITC Annexin V apoptosis detection kit and PI (Bio Legend) were used according to the manufacturer’s instructions. All samples were immediately analyzed using an FACS Calibur flow cytometry (Beckton Dickinson, Franklin Lakes, NJ, USA).

### 2.8. Protein Extraction and Western Blotting Analysis

Proteins were extracted using the radioimmunoprecipitation assay buffer (Biosesang, Seongnam, Republic of Korea) supplemented with a protease inhibitor (Invitrogen, Waltham, MA, USA). Protein concentration was measured using the DC protein assay kit (Bio-Rad, Hercules, CA, USA) and adjusted to 20 µg per well for sodium dodecyl sulfate polyacrylamide-gel electrophoresis. All samples were mixed with the sample buffer (Bio-Rad), heated on a 95 °C heating block for 5 min, and stored at −20 °C. The protein samples were separated on a 12% acrylamide gel using electrophoresis and transferred onto a polyvinylidene fluoride membrane (Bio-Rad). The membrane was blocked with 5% skim milk in TBST for 1.5 h and incubated with primary antibodies at 4 °C overnight, followed by incubation with suitable secondary antibodies at room temperature for 1.5 h. Protein bands were visualized using the ECL Kit (Thermo Fisher Scientific, San Jose, CA, USA), and iBright CL1000 (Thermo Fisher Scientific, Waltham, MA, USA) was used for quantification. The following primary antibodies were used: caspase 3 (1:1000; Novus Bio, Centennial, CO, USA), Bcl-2-associated X (BAX; 1:1000; Santa Cruz Biotechnology, Dallas, TX, USA), heat shock protein (HSP)-90 (1:1000; Abcam, Cambridge, UK), HSP70 (1:1000; Enzo, Farmingdale, NY, USA), HSP60 (1:1000; Enzo), HSP47 (1:1000; Enzo), HSP40 (1:1000; Enzo), and glyceraldehyde 3-phosphate dehydrogenase (GAPDH; 1:5000; Invitrogen). Goat anti-mouse IgG (Thermo Fisher Scientific, New York, NY, USA) and goat anti-rabbit IgG (Thermo Fisher Scientific, New York, NY, USA) were the secondary antibodies used here. All protein bands were normalized to those of GAPDH as an internal control, and the protein expression levels were expressed as relative values to GAPDH.

### 2.9. Statistical Analyses

All statistical analyses were conducted using the SAS version 9.4 software (SAS Institute Inc., Cary, NC, USA). Analysis of variance followed by Student’s *t*-test were used to compared the CopA3-treated and untreated groups. Duncan’s multiple-range test was used for all other comparisons, as appropriate. Data are represented as the mean ± standard error. Statistical significance was set at *p* < 0.01 and *p* < 0.05.

## 3. Results

### 3.1. CopA3 Cytotoxicity on PMSCs

We conducted immunocytochemistry to confirm the presence of satellite cells (Figure 1A). PAX7 has been recognized as a specific marker in PMSCs [22,23], and MYOD is classified as one of the myogenic regulatory factors that can co-express with PAX7 [23,24]. The percentage of PAX7+/DAPI+ cells was 95.9 ± 0.7% in PMSCs, whereas it was 4.2 ± 0.7% in fibroblasts (*p* < 0.01) (Figure 1B). In addition, the percentage of MYOD-positive cells was significantly higher (*p* < 0.01) in PMSCs than in fibroblasts (Figure 1C). We measured the cytotoxicity of various concentrations of CopA3 on PMSCs (Figure 1). Cell survival rate was significantly higher with 10 µg/mL of CopA3 than with the other doses. However, the survival rate decreased in a dose-dependent manner at 25 and 50 µg/mL. Based on this result, we selected 10 µg/mL of CopA3 for subsequent analysis.

### 3.2. Effect of CopA3 on PMSC Viability

We determined the effect of CopA3 on the viability of PMSCs under heat stress. No morphological changes were observed in PMSCs at different temperatures and CopA3 treatment (Figure 2A). However, as the incubation temperature increased, cell viability significantly decreased (*p* < 0.01) in CopA3-untreated groups (Figure 2B). Interestingly, CopA3 treatment significantly improved the cell viability compared to that of the control (37 °C, CopA3^(−)^) group at all temperatures (*p* < 0.01; Figure 2B).

### 3.3. Effect of CopA3 on the Cell Cycle Distribution of PMSCs

To examine the effect of CopA3 on the PMSC cell cycle, we analyzed the cell cycle distribution at 37, 39, and 41 °C (Figure 3). The proportion of cells in the G0/G1 phase increased with increasing temperature (*p* < 0.01) (Figure 3A,B). However, CopA3 significantly reduced (*p* < 0.01) the proportion of cells in the G0/G1 phase at all temperatures (Figure 3A,B). The proportion of cells in the S and G2/M phases also decreased with increasing temperature; however, CopA3 effectively reversed this decreased effect (Figure 3A,C,D).

### 3.4. Effect of CopA3 on PMSC Apoptosis

Next, we investigated the effect of CopA3 on PMSC apoptosis (Figure 4). The ratio of live cells decreased as the temperature increased in both the CopA3-treated and untreated groups (*p* < 0.01) (Figure 4A,B). However, CopA3 treatment significantly increased the ratio of live cells compared to that in the untreated groups at all temperatures in this study (Figure 4A,B). In addition, heat stress increased (*p* < 0.05) the apoptosis of PMSCs, whereas CopA3 treatment significantly decreased (*p* < 0.05) 37 and 41 °C (Figure 4A,C). Next, we verified whether the cytoprotective effect of CopA3 was due to its regulatory effects on apoptosis-related proteins, such as caspase 3 and BAX (Figure 4D–F). Expression levels of caspase 3, an apoptosis-inducing protein, were not significantly different among all groups (Figure 4D,E). However, expression levels of the pro-apoptotic protein, BAX, significantly increased (*p* < 0.05) with increasing temperatures in CopA3-untreated groups (Figure 4D,F). CopA3 restored BAX expression to a level comparable to that in the control group at all temperatures (Figure 4D,F). Notably, BAX expression levels were remarkably decreased (*p* < 0.05) after CopA3 treatment at 41 °C in PMSCs (Figure 4D,F).

### 3.5. Changes in the Expression Levels of HSPs in PMSCs

To verify the role of CopA3 in the regulation of HSP expression under heat stress, we investigated the expression levels of HSPs. HSP90 levels were significantly increased at 39 and 41 °C in both the CopA3-treated (*p* < 0.05) and untreated (*p* < 0.01) groups (Figure 5A,B). HSP70 levels were significantly increased (*p* < 0.05) with increasing temperature in the CopA3-untreated groups. However, no significant differences in HSP70 expression levels were observed among the CopA3-treated groups at different temperatures (Figure 5A,B). Expression levels of HSP60 and HSP47 did not differ significantly among the groups (Figure 5A,B). Similar to HSP70, HSP40 levels significantly increased (*p* < 0.05) with temperature in the CopA3-untreated groups but did not change in the CopA3-treated groups. In particular, HSP40 levels were significantly decreased (*p* < 0.05) after the CopA3 treatment of PMSCs at 41 °C (Figure 5A,B).

## 4. Discussion

Pigs are highly susceptible to heat stress due to factors such as limited sweat glands and a high metabolic rate [22]. Considering that heat stress affects not only feed intake but also protein refolding and energy utilization in the body [22], conducting the research on materials to mitigate heat stress becomes crucial. However, there have been no studies demonstrating the heat-stress-alleviating effects of the insect peptide CopA3 in PMSCs. In this study, we investigated the impact of adding CopA3 (10 μg/mL) to the culture medium on the viability of heat-stressed PMSCs. Cell viability is known to depend on temperature [23]. Consist with our previous report [13], we observed a decrease in cell viability as the temperature increased in this study, suggesting that treatment with CopA3 restored the heat stress-induced reduction in cell viability.

Cell proliferation is controlled by the cell cycle and apoptosis [24]. Our results align with previous reports indicating that heat stress inhibits cell proliferation via temporary arrest at the cell cycle checkpoints of G1/S and G2/M transitions [25]. Notably, S phase of the cell cycle is particularly sensitive to cell proliferation and death [26]. Various types of stimulation can disrupt the duration of different cell cycle phases, including DNA synthesis in the S phase, and can potentially cause damage to cellular DNA. S and G2/M phases together represent the cell proliferation stage [27]. Here, the addition of CopA3 increased cell viability compared to the untreated group at 41 °C, primarily due to the increase in S- and G2/M-phase cell populations. These results suggest that CopA3 promotes DNA synthesis and increases the population of cells in S and G2/M phases under heat stress.

Heat stress induces changes in cellular proteins and causes cell death [18]. We previously reported a decrease in live cells and increase in apoptosis under heat stress conditions at 39 and 41 °C [13]. The results of this study also align with our previous observation. In addition, CopA3 treatment reduced the expression of the anti-apoptotic protein, BAX, which was increased by heat stress. However, both the heat-stress conditions (39 and 41 °C) and CopA3 treatment had no significant effect on caspase 3 expression. Kim et al. [28] reported that CopA3 inhibits apoptosis in hepatocytes under various apoptosis-inducing conditions, such as the exposure to ethanol or hydrogen peroxide and serum starvation, by blocking caspase 3 and caspase 6. However, whether the alleviation of heat stress-induced apoptosis is specifically due to the inhibitory effect of CopA3 on caspase 3 remains unknown. BAX primarily resides in the cytoplasm in an inactive state [29,30]. Upon apoptotic stimulation, BAX translocates to the mitochondria and becomes active [29,30,31]. CopA3 exerts its effects by localizing in the cytosolic fraction. Caspase 3 is also predominantly located in the cytoplasm [15]. Here, we found that CopA3 reduced the apoptosis of PMSCs, possibly via regulating BAX expression. Further studies are necessary to elucidate the effects of CopA3 on other apoptosis-related proteins and its underlying action mechanisms.

Various cell types reduce heat stress by upregulating the expression of the molecular chaperones, HSPs [32]. Under heat stress, heat shock factors are translocated from the cytoplasm to the nucleus, where they bind to heat shock elements to synthesize HSPs that protect the cell by preventing protein aggregation, misfolding, and unfolding [33,34]. In addition, HSPs expressed in response to heat stress are directly involved in myogenesis [35]. HSP70 regulates muscle regeneration, prevents apoptosis, and exhibits higher expression in differentiated myotubes than in proliferating myoblasts [35,36,37] HSP90, and HSP60 plays a crucial role in myoblast survival by preventing apoptosis and protein aggregation [35,38,39,40]. HSP47 is an important molecular chaperone that facilitates collagen synthesis and maturation [41]. HSP40 regulates the cell cycle changes induced by heat stress in myoblasts [42]. In addition, HSP40 interacts with HSP70 and acts as a co-chaperone [43,44,45]. HSP70 and HSP40 are distributed throughout the cytoplasm and nucleus, whereas HSP60 is localized in the perinuclear space of mouse myoblasts [35]. As previously mentioned, CopA3 is known to function in the cytoplasm. Hence, it is postulated that the selective response of CopA3 to these HSPs is closely linked to the distribution location of each factor. Further research is deemed necessary on this; nevertheless, our findings suggest that HSP70 and HSP40 function collaboratively in PMSCs under heat stress and could be stabilized by CopA3.

Betaine, as a heat stress reliever, has been demonstrated to decrease intracellular apoptosis [46] and reduce the upregulation of HSPs by stabilizing intracellular proteins, thus alleviating cellular stress [47]. Moreover, the provision of a betaine-supplemented diet resulted in an increase in the size of muscle fibers in pigs, thereby facilitating skeletal muscle growth [48]. Selenium also has been shown to reduce apoptosis caused by heat stress, thereby reducing the need for cell to synthesize HSP70 protein as a response to heat stress in intestinal cells [22,49]. Furthermore, selenium supplementation restored the expression of Myogenin, a muscle differentiation marker that was downregulated under heat-stress conditions, to the level observed in non-heat-stressed mouse myoblasts [50]. Taurine, which is recognized as a natural antioxidant found in various animal tissues [51], has also been investigated as a potential remedy for heat stress [52]. Taurine alleviated the rates of apoptosis in bovine mammary epithelial cells and suppressed the expression of heat shock factor 1 (HSF1) mRNA and protein under heat-stress conditions, but not HSP90 [53]. Our findings align with the research conducted by Bai, Li, Yu, Zhou, Kou, Guo, Yang and Yan [53], as we also observed selective response of CopA3 towards HSPs. The compounds under investigation as heat-stress relievers have shown the capability to restore apoptosis and protein expression levels, which were increased by heat stress, back to normal conditions. In this study, CopA3 also decreased the apoptosis in PMSCs while enhancing the stability of HSPs under heat-stress conditions. These findings support the potential application of CopA3 as a promising mitigating agent for heat stress.

## 5. Conclusions

In this study, we found that 10 μg/mL of CopA3 increased PMSC proliferation by promoting the transition from the G1 to S phase and maintaining stable HSP expression at elevated temperatures. Moreover, CopA3 alleviated heat stress by increasing the proliferation and decreasing the apoptosis of PMSCs under heat stress. It also stabilized the expression levels of BAX, HSP70, and HSP40, thereby preventing cell death. Our findings suggest that CopA3 has the potential to improve animal productivity in the pork industry as a heat-stress alleviation compound. However, further studies are necessary to elucidate its precise action mechanisms.

## Figures and Tables

**Figure 1 animals-13-03209-f001:**
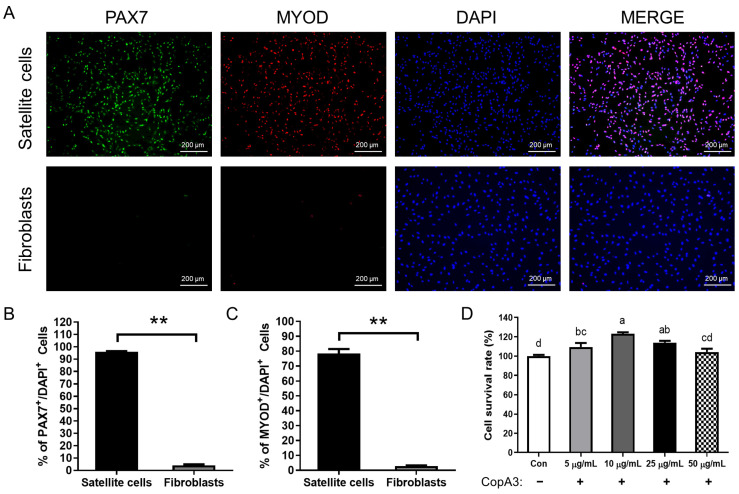
Verification of porcine muscle satellite cells (PMSCs) and cell survival at different CopA3 doses. (**A**) Immunostaining images of PMSCs and fibroblast cells. Cell nuclei were stained with PAX7 (green), MYOD (red), and DAPI (blue). (**B**,**C**) Quantification of the percent of PAX7^+^ and MYOD^+^ PMSCs and fibroblasts. *n* = 3. Student’s *t*-test was employed to assess significant differences between the satellite cells and fibroblasts. ** *p* < 0.01. (**D**) The CCK-8 assay was performed to determine the optimal concentration of CopA3 for treatment. *n* = 5. All values are represented as the mean ± standard error (SE). ^a–d^ Different letters indicate the statistically significant differences (*p* < 0.01), and same letters (ex: a and ab) indicates no significant differences among the treatment groups.

**Figure 2 animals-13-03209-f002:**
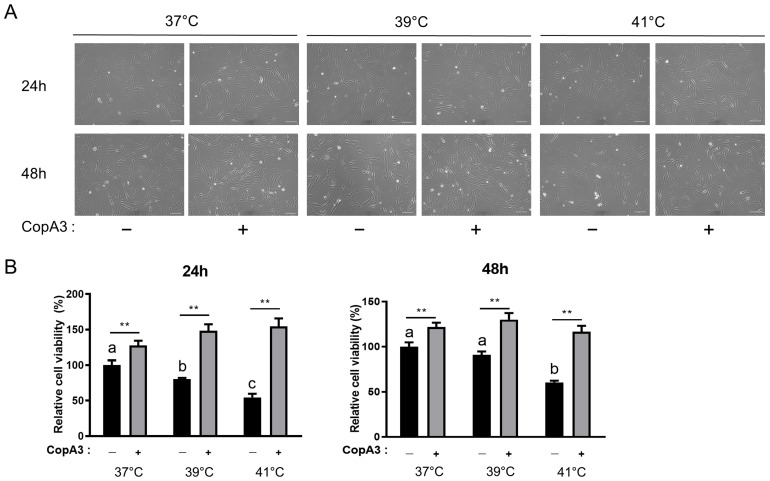
Proliferation of PMSCs under heat stress CopA3 treatment. (**A**) Microscopy image of PMSCs incubated under various temperatures with CopA3 for 48 h. Scale bar = 100 µm. (**B**) Cell viability was measured using the cell counting kit (CCK)-8 assay. *n* = 10. Values are represented as the mean ± SE. The Duncan multiple--range test was used to compare temperature-related differences within groups treated with 10 µg/mL of CopA3^(+)^ and CopA3^(−)^ groups, respectively. ^a–c^ Different letters denote statistically significant differences among the groups (*p* < 0.01) in Duncan multiple rage test. Student’s *t*-test was employed to assess significant differences between the two groups treated with 10 µg/mL of CopA3^(+)^ and CopA3^(−)^ groups at the same temperature. ** *p* < 0.01.

**Figure 3 animals-13-03209-f003:**
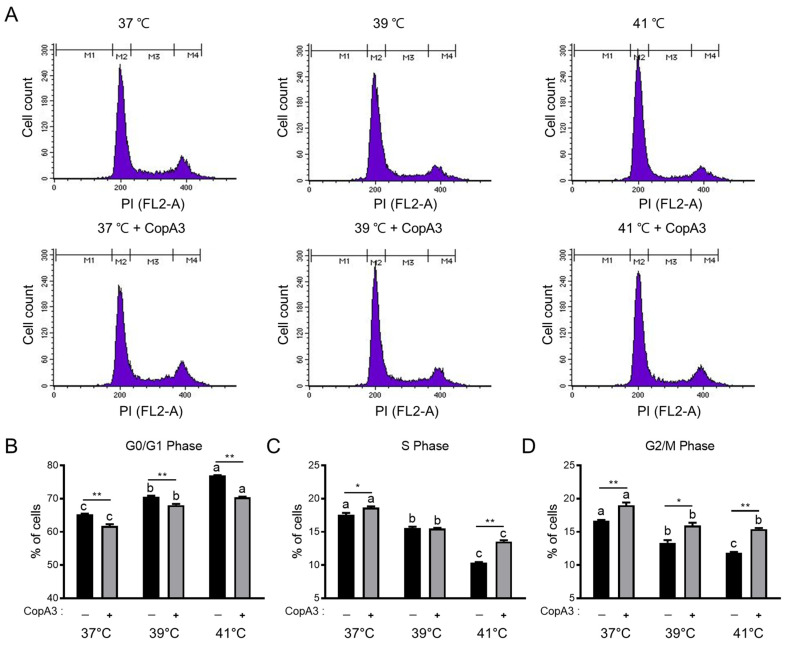
Flow cytometry analysis of the cell cycle phase distribution in PMSCs. (**A**) Schematic diagram of cell cycle analysis results via propidium iodide (PI) staining. (**B**) Percentages of PMSCs in the G0/G1, (**C**) S, and (**D**) G2/M phase. Values are represented as the mean ± SE. *n* = 3. The Duncan multiple range test was used to compare temperature-related differences within groups treated with 10 µg/mL of CopA3^(+)^ and CopA3^(–)^ groups, respectively. ^a–c^ Different letters denote statistically significant differences among the groups (*p* < 0.01) in Duncan’s multiple-range test. Student’s *t*-test was employed to assess significant differences between the two groups treated with 10 µg/mL of CopA3^(+)^ and CopA3^(–)^ groups at the same temperature. ** *p* < 0.01 and * *p* < 0.05.

**Figure 4 animals-13-03209-f004:**
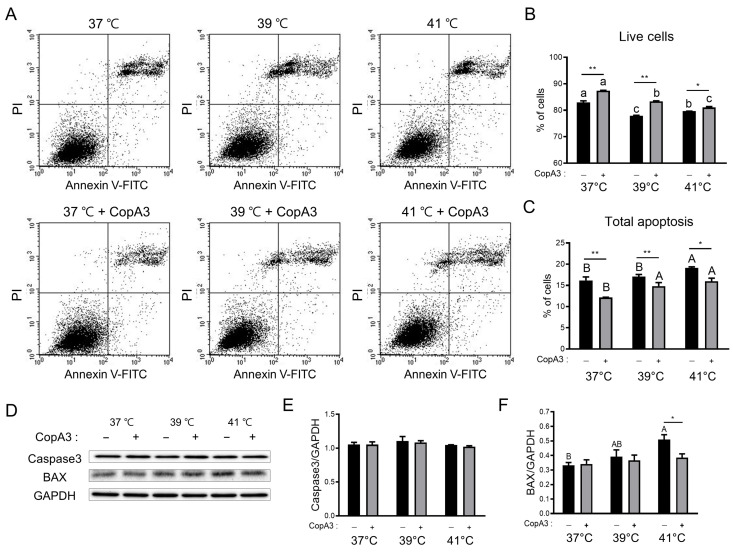
The addition of CopA3 decreases cell apoptosis at all temperatures. (**A**) Dot plot of apoptosis analysis via flow cytometry. (**B**) Percentages of live cells and (**C**) total apoptosis. (**D**) Comparison of apoptosis-related protein bands with or without CopA3 treatment at different temperatures (37, 39, and 41 °C). (**E**,**F**) Protein expression levels were normalized using glyceraldehyde 3-phosphate dehydrogenase (GAPDH) levels. Data are represented as the mean ± SE. *n* = 3. The Duncan multiple-range test was used to compare temperature-related differences within groups treated with 10 µg/mL of CopA3^(+)^ and CopA3^(–)^ groups, respectively. ^a–c^ Different letters denote statistically significant differences among the groups (*p* < 0.01) in Duncan multiple rage test. ^A–B^ Different letters indicate the statistically significant differences among treatment groups (*p* < 0.05) in the Duncan multiple-range test. Student’s *t*-test was employed to assess significant differences between the two groups treated with 10 µg/mL of CopA3^(+)^ and CopA3^(–)^ groups at the same temperature. ** *p* < 0.01 and * *p* < 0.05.

**Figure 5 animals-13-03209-f005:**
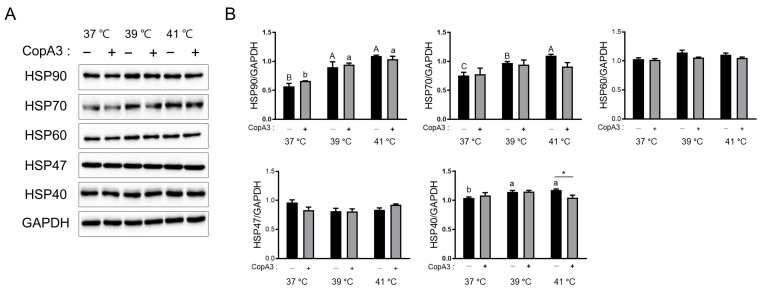
Heat shock protein (HSP) expression levels with or without CopA3 treatment at different temperatures. (**A**) Comparison of HSP bands using Western blotting. (**B**) HSP expression levels were normalized using GAPDH levels. *n* = 3. Values are represented as the mean ± SE. The Duncan multiple-range test was used to compare temperature-related differences within groups treated with 10 µg/mL of CopA3^(+)^ and CopA3^(–)^ groups, respectively. ^a–b^ Different letters denote statistically significant differences among the groups (*p* < 0.01) in the Duncan multiple-range test. ^A–C^ Different letters indicate the statistically significant differences among treatment groups (*p* < 0.05) in the Duncan multiple-range test. Student’s *t*-test was employed to assess significant differences between the two groups treated with 10 µg/mL of CopA3^(+)^ and CopA3^(–)^ groups at the same temperature. * *p* < 0.05.

## Data Availability

Not applicable.

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
