# Peer review of "Insect Peptide CopA3 Mitigates the Effects of Heat Stress on Porcine Muscle Satellite Cells"

_animals, 2023, doi:10.3390/ani13203209_

Round 1
Reviewer 1 Report
The authors applied in vitro insect peptide CopA3 to mitigate the effects of heat stress on porcine muscle satellite cells isolated.
The study has merit as uses flow cytometry and Western blotting to give evidence. The manuscript is well written and clear. It is unclear though which cells were sorted and treated. Authors did not add information on cell sorting strategy, % of purity and markers used in the sorting. The graph presentation needs to be improved to a publishable quality. Further detail on the suggestions is described below.
MAJOR COMMENTS
Methods
1. How do authors know they have isolated satellite cells? The described method will enable isolation of adherent cells including fibroblasts. Add method on how cells were sorted, and cell markers.
Results
1. What percentage are satellite cells and fibroblasts? Add graph demonstrating the sorting strategy.
2. Figures do not need extra title added at the top of the graphs. It can be removed.
3. Figure 1 statical labelling is not clear. The format using letters is not appropriate and does not make clear comparisons. Add detailed information in figure legend.
Add single point samples in the bar graph to clarify the statistical significance p<0.01
Improve graph presentation to a higher academic standard. Units presentation, X-axis, statistics
4. Figure 2. Revise format. Remove repetitive information, the format used is confusing and does not clarify data. What a,b, c means. Figure legend needs to be written in a clearer way.
5. Figure 3. Clarify the marker used in flow cytometry measurement of G0/G1, S, G2/M phases.
MINOR COMMENTS
Revise and edit text, all symbols should be defined before use.
Line 103 Grammar
The manuscript is clear and well structured. Revise text for the use of symbols as they need to be defined when appearing at the first time in the text. There are few improvements with grammar needed.
Reviewer 2 Report
Summary:
The authors evaluate the effectiveness of utilizing the insect-derived peptide CopA3 to modulate the proliferation and heat shock protein (HSP) expression in porcine muscle satellite cells (PMSCs). The study reveals that CopA3 administration ameliorates the adverse effects of heat stress on cellular viability. Additionally, CopA3 can attenuate apoptosis in PMSCs while stabilizing Bax, HSP70, and HSP40 expression levels. These findings hold significant implications for developing compounds to alleviate heat stress. The manuscript is fundamentally robust; however, it is recommended that certain modifications be undertaken for further refinement.
Major comments:
1. The author mentions, "These results have implications for the advance of heat stress-alleviating compounds." While the discussion section does touch upon some compounds, further elaboration on previously discovered heat stress-alleviating compounds would enhance the paper's comprehensiveness.
2. In Figure 1, the alphabetical notation ranging from "a" to "d" is utilized to signify statistically significant differences among the various treatment groups. A comprehensive elucidation of the specific meaning attributed to each letter is strongly recommended. Analogous clarifications are suggested for other figures, such as Figure 4, where uppercase "A" and lowercase "a" denote statistical differences.
3. Figure 2A presents microscopic images of PMSCs at the 48-hour time point. Including observations at the 24-hour would offer valuable insights into potential morphological alterations.
4. In Figure 4D, the clarity of the Western blot corresponding to Bax is compromised, raising concerns about the accuracy of subsequent quantification analyses. It is suggested that the blot be re-run to ensure reliable data interpretation.
5. In Figures 4 and 5, Western blot quantification methodologies are employed; however, the approach for normalizing to GAPDH is not explicitly delineated in the Materials and Methods section. The provision of this crucial information is strongly encouraged.
Minor comments:
1. In line 66, the phrase "proliferation of heat-stressed PMSCs in vitro" appears to be incomplete or lacking context. Additionally, "in vitro" should be italicized for proper formatting.
2. The abbreviation "AMPs" is used in line 62; please provide its full name for clarity.
3. In Figures 2 and 3, the CopA3-treated group is specified to have been treated with a concentration of 10 μg/mL. If the same concentration is used in Figures 4 and 5, this should be indicated in the figure legends.
4. There is an inconsistency in the units used for concentration; Figure 1 uses "10 μg/ml," while line 157 uses "10 μg/mL." Please standardize the format.
5. Lastly, Figure 5 does not include the notation "**", so it is advisable to refrain from mentioning "** p<0.01" in the text.
Moderate editing of English language required.
Author Response
please see the attachmant.

Round 2
Reviewer 2 Report
There are inconsistencies in the units used for concentration. For example, Line 202 uses "10 μg/ml," while Line 205 uses "10 μg/mL." Please check the whole paper and standardize the format.
Moderate editing of English language required
Author Response
There are inconsistencies in the units used for concentration. For example, Line 202 uses "10 μg/ml," while Line 205 uses "10 μg/mL." Please check the whole paper and standardize the format.
ANS: Thank you for your keen observation. We unified the format in our revised manuscript.